# Fuzzy c-Means Clustering in Persistence Diagram Space for Deep Learning Model Selection

**Thomas Davies**                                    T.O.M.DAVIES@SOTON.AC.UK
*University of Southampton, UK.*

**Jack Aspinall**                                    JACK.ASPINALL@ORIEL.OX.AC.UK
*University of Oxford, UK.*

**Bryan Wilder**                                    BWILDER@ANDREW.CMU.EDU
*Carnegie Mellon University, US.*

**Long Tran-Thanh**                                    LONG.TRAN-THANH@WARWICK.AC.UK
*University of Warwick, UK.*

**Editors:** Sophia Sanborn, Christian Shewmake, Simone Azeglio, Arianna Di Bernardo, Nina Miolane

## Abstract

Persistence diagrams concisely capture the structure of data, an ability that is increasingly being used in the nascent field of topological machine learning. We extend the ubiquitous Fuzzy c-Means (FCM) clustering algorithm to the space of persistence diagrams, enabling unsupervised learning in a topological setting. We give theoretical convergence guarantees that correspond to the Euclidean case and empirically demonstrate the capability of the clustering to capture topological information via the fuzzy RAND index. We present an application of our algorithm to a scenario that utilises both the topological and fuzzy nature of our algorithm: pre-trained model selection in deep learning. As pre-trained models can perform well on multiple tasks, selecting the best model is a naturally fuzzy problem; we show that fuzzy clustering persistence diagrams allows for unsupervised model selection using just the topology of their decision boundaries.

**Keywords:** Topological Data Analysis, Fuzzy Clustering, Model Selection

## 1. Introduction

Persistence diagrams, a concise representation of the topology of a point cloud with strong theoretical guarantees, have emerged as a new tool in the field of data analysis (Edelsbrunner and Harer, 2010). Persistence diagrams have been successfully used to analyse problems ranging from identifying financial crashes (Gidea and Katz, 2018) to analysing protein bindings (Kovacev-Nikolic et al., 2014), In this paper, we contribute to the increasing number of topological machine learning techniques by extending the Fuzzy c-Means (FCM) clustering algorithm (Bezdek, 1980) to the space of persistence diagrams. It is widely accepted that many real-world datasets are not clearly delineated into hard categories (Campello, 2007). Thus any algorithm that accounts for this is desirable, as evidenced by the large number of publications studying and using fuzzy clustering algorithms (Li and Lewis, 2016; Yang et al., 2019; Pantula et al., 2020). Our algorithm enables practitioners to study the fuzzy nature of data through a topological lens directly in the space of persistence diagrams.

We perform the convergence analysis for our algorithm, giving the same guarantees as traditional FCM clustering: that every convergent subsequence of iterates tends to a local

minimum or saddle point. As this guarantee could lead to non-convergence in practice, we empirically evaluate convergence on a total of 825 randomly generated persistence diagrams and find that the algorithm converges every time. We evaluate the algorithm using a variety of distances on persistence diagrams with the fuzzy RAND index (Campello, 2007), a standard measure of cluster quality. We find that we fall into the standard paradigm: distances that take longer to compute result in higher quality clustering, and approximating distances leads to lower quality clusters that can be computed faster.

We demonstrate the practical value of fuzzy clustering persistence diagrams with an application to pre-trained model selection. This application is inspired by the somewhat surprising previous work showing that pre-trained deep learning models perform better on tasks if they have topologically similar decision boundaries (Ramamurthy et al., 2019). As one model can perform well on multiple tasks, this is a naturally fuzzy problem, and so ideally suited to our algorithm. We use our method to cluster pre-trained models and tasks (unseen datasets) using only the persistence diagrams of their decision boundaries. Not only is our algorithm able to successfully cluster models to the task it's originally trained on based just on the topology of its decision boundary, but we demonstrate that higher fuzzy cluster membership values imply better performance on tasks that the model has not been trained on.

## 1.1. Related work

**Means of persistence diagrams.** Before we can cluster in the space of peristence diagrams we need to be able to compute means. Mileyko et al. (2011) first showed that means and expectations are well-defined in the space of persistence diagrams. Specifically, they showed that the Fréchet mean, an extension of means onto metric spaces, is well-defined under weak assumptions on the space of persistence diagrams. Turner et al. (2012) then developed an algorithm to compute the Fréchet mean. However, the combinatoric nature of their algorithm makes it computationally intense. There is a relevant line of research for speeding up the computation of means and barycentres. In particular, Lacombe et al. (2018) framed the computation of means and barycentres in the space of persistence diagram as an optimal transport problem, allowing them to use the Sinkhorn algorithm Cuturi and Doucet (2014) for fast computation of approximate solutions. Techniques to speed up the underlying matching problem fundamental to our computation have also been proposed by Vidal et al. (2020) and Kerber et al. (2017). Our fuzzy clustering algorithm can integrate these solutions to further speed up its computing time if necessary.

**Learning with persistence-based summaries.** Integrating diagrams into machine learning workflows remained challenging even with well-defined means, as the space is non-Hilbertian (Turner and Spreemann, 2019). As such, efforts have been made to map diagrams into a Hilbert space; primarily either by embedding into finite feature vectors (Kališnik, 2018; Fabio and Ferri, 2015; Chepushtanova et al., 2015) or functional summaries (Bubenik, 2015; Rieck et al., 2019), or by defining a positive-definite kernel on diagram space (Reininghaus et al., 2015; Carrière et al., 2017; Le and Yamada, 2018). These vectorisations have been integrated into deep learning either by learning parameters for the embedding (Hofer et al., 2017; Carrière et al., 2020; Kim et al., 2020; Zhao and Wang, 2019; Zieliński et al., 2019), or as part of a topological loss or regulariser (Chen et al., 2018; Gabrielsson

et al., 2020; Clough et al., 2020; Moor et al., 2019). However, the embeddings used in these techniques deform the metric structure of persistence diagram space (Bubenik and Wagner, 2019; Wagner, 2019; Carrière and Bauer, 2019), potentially leading to the loss of important information. In comparison, our algorithm acts in the space of persistence diagrams so it does not deform the structure of diagram space via embeddings. However, as an unsupervised learning algorithm, our algorithm is intended to complement these techniques, offering a different approach for practitioners to use, rather than directly competing with them.

Maroulas et al. (2017) gave an algorithm for hard clustering persistence diagrams based on the algorithm by Turner et al. (2012). As mentioned earlier, many real-world datasets are not clearly delineated into hard categories (Campello, 2007), and a fuzzy algorithm would naturally be chosen over a hard clustering algorithm when dealing with such datasets.

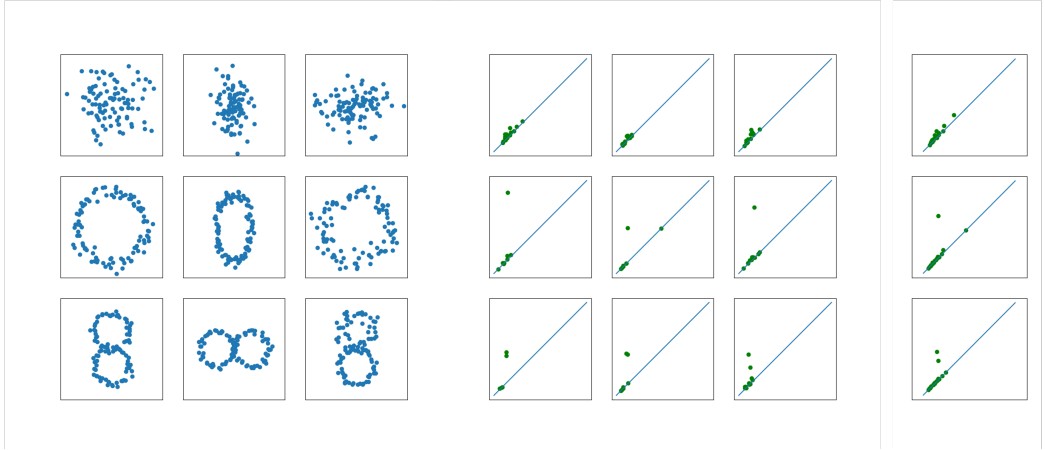

Figure 1: On the left we show nine synthetic datasets, consisting of noise, one hole, or two holes. In the middle we compute the 1-persistence diagrams, which we recall counts the number of holes. We cluster these persistence diagrams, resulting in three learnt cluster centres, shown on the right. The cluster centres have zero, one, and two significantly off-diagonal points: the clustering algorithm has learnt the topological features of the datasets.

## 2. Topological preliminaries

Topological Data Analysis emerged from the study of algebraic topology, providing a toolkit to fully describe the topology of a dataset. We offer a quick summary below; for more comprehensive details see Edelsbrunner and Harer (2010). A set of points in $\mathbb{R}^d$ are indicative of the shape of the distribution they are sampled from. By connecting points that are pairwise within $\epsilon > 0$ distance of each other, we can create an approximation of the distribution called the Vietoris-Rips complex (Vietoris, 1927). Specifically, we add the convex hull of any collection of points that are pairwise at most $\epsilon$ apart to the $\epsilon$-Vietoris-Rips complex. However, choosing an $\epsilon$ remains problematic; too low a value and key points can remain disconnected, too high a value and the points become fully connected. To overcome this

we use *persistence*: we consider the approximation over all values of $\epsilon$ simultaneously, and study how the topology of that approximation evolves as $\epsilon$ grows. We call the collection of complexes for all $\epsilon$ a filtration.

For each $\epsilon$, we compute the $p$-homology group. This tells us the topology of the $\epsilon$-Vietoris-Rips complex: the 0-homology counts the number of connected components, the 1-homology counts the number of holes, the 2-homology counts the number of voids, and so on (Edelsbrunner et al., 2000). The $p$-persistent homology group is created by summing the $p$-homology groups over all $\epsilon$. This results in a $p$-PH group that summarises information about the topology of the dataset at all granularities. If a topological feature only persists for a short amount of time, then it's more likely to be noise (Cohen-Steiner et al., 2007). We can stably map a $p$-PH group into a multiset in the extended plane called a persistence diagram (Chazal et al., 2012). Each topological feature has a birth and death – a feature is 'born' when it enters the filtration and 'dies' when it is destroyed. The birth and death values (i.e., the values of $\epsilon$ when a topological feature enters the filtration or is destroyed) are the axes of the persistence diagram, so each point in the persistence diagram represents a topological feature. The larger the difference between birth and death values, the longer a topological feature *persists* for, and the more likely the feature is to be a feature of the distribution that the points are sampled from. By computing the birth and death points for each topological feature in the filtration, we get a complete picture of the topology of the point cloud at all granularities (Zomorodian and Carlsson, 2005). The persistence diagram is the collection of birth/death points, along with the diagonal $\Delta = \{(a, a) : a \in \mathbb{R}\}$ with infinite multiplicity, added in order to make the space of persistence diagrams complete (Mileyko et al., 2011).

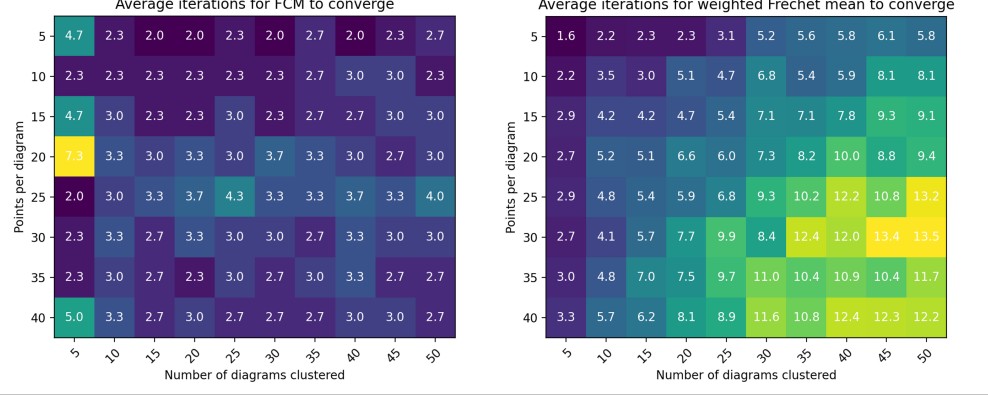

Figure 2: Heatmaps showing average number of iterations for fuzzy clustering of persistence diagrams (left) and the weighted Fréchet mean computation (right) to converge. Convergence of the FCM algorithm is determined when the cost function is stable to within ±0.5%. Convergence experiments were carried out on a total of 825 persistence diagrams, including three repeats .

## 3. Algorithmic design

### 3.1. Clustering persistence diagrams

In order to cluster we need a distance on the space of persistence diagrams. We use the 2-Wasserstein $L_2$ metric as it is stable on persistence diagrams of finite point clouds (Chazal et al., 2012). The Wasserstein distance is an optimal transport metric that has found applications across machine learning. In the Euclidean case, it quantifies the smallest distance between optimally matched points. Given diagrams $\mathbb{D}_1, \mathbb{D}_2$, the distance is

$$W_2(\mathbb{D}_1, \mathbb{D}_2) = \left( \inf_{\gamma:\mathbb{D}_1 \to \mathbb{D}_2} \sum_{x \in \mathbb{D}_1} ||x - \gamma(x)||_2^2 \right)^{1/2},$$

where the infimum is taken over all bijections $\gamma : \mathbb{D}_1 \to \mathbb{D}_2$. Note that as we added the diagonal with infinite multiplicity to each diagram, these bijections exist. If an off-diagonal point is matched to the diagonal the transportation cost is simply the shortest distance to the diagonal. In fact, the closer a point is to the diagonal, the more likely it is to be noise (Cohen-Steiner et al., 2007), so this ensures our distance is not overly affected by noise.

We work in the space $\mathscr{D}_{L^2} = \{\mathbb{D} : W_2(\mathbb{D}, \Delta) < \infty\}$,[1] as this leads to a geodesic space with known structure (Turner et al., 2012). Given a collection of persistence diagrams $\{\mathbb{D}_j\}_{j=1}^n \subset \mathscr{D}_{L^2}$ and a fixed number of clusters $c$, we wish to find cluster centres $\{\mathbb{M}_k\}_{k=1}^c \subset \mathscr{D}_{L^2}$, along with membership values $r_{jk} \in [0,1]$ that denote the extent to which $\mathbb{D}_j$ is associated with cluster $\mathbb{M}_k$. We follow probabilistic fuzzy clustering, so that $\sum_k r_{jk} = 1$ for each $j$.

We extend the FCM algorithm originally proposed by Bezdek (1980). Our $r_{jk}$ is the same as traditional FCM clustering, adapted with the Wasserstein distance. That is,

$$r_{jk} = \left( \sum_{l=1}^c \frac{W_2(\mathbb{M}_k, \mathbb{D}_j)}{W_2(\mathbb{M}_l, \mathbb{D}_j)} \right)^{-1}. \tag{1}$$

To update $\mathbb{M}_k$, we compute the weighted Fréchet mean $\hat{\mathbb{D}}$ of the persistence diagrams $\{\mathbb{D}_j\}_{j=1}^n$ with the weights $\{r_{jk}^2\}_{j=1}^n$.

Specifically,

$$\mathbb{M}_k \longleftarrow \arg\min_{\hat{\mathbb{D}}} \sum_{j=1}^n r_{jk}^2 W_2(\hat{\mathbb{D}}, \mathbb{D}_j)^2, \text{ for } k = 1, \ldots, c. \tag{2}$$

As the weighted Fréchet mean extends weighted centroids to general metric spaces, this gives our fuzzy cluster centres. The computation of the weighted Fréchet mean is covered in Section 3.2. By alternatively updating (1) and (2) we get a sequence of iterates. Theorem 1, proven in Appendix A, provides the same convergence guarantees as traditional FCM clustering.

---

1. To ensure that our persistence diagrams are all in this space, we map points at infinity to a hyper-parameter $T$ that is much larger than other death values in the diagram. Alternatively, this can be avoided entirely by computing the diagrams with extended persistence (Cohen-Steiner et al., 2009), which removes points at infinity.

THEOREM 1:
Every convergent subsequence of the sequence of iterates obtained by alternatively updating membership values and cluster centres with (1) and (2) tends to a local minimum or saddle point of the cost function $J(R, \mathbb{M}) = \sum_{j=1}^{n} \sum_{k=1}^{c} r_{jk}^2 W_2(\mathbb{M}_k, \mathbb{D}_j)^2$.

Observe that we only guarantee the convergence of subsequences of iterates. This is the same as traditional FCM clustering, so we follow the same approach to a stopping condition and run our algorithm for a fixed number of iterations.

### 3.2. Computing the weighted Fréchet mean

Turner et al. (2012) give an algorithm for the computation of Fréchet means. In this section we extend their algorithm and proof of convergence to the weighted case. The proof is by gradient descent, which requires defining a differential structure on the space of persistence diagrams. Our extension to the proof comes down to proving that given some supporting vectors of the Fréchet function, the weighted sum of those is also a supporting vector. For more details see Appendix B.

To give some intuition, start by recalling that when processing the persistence diagrams we add copies of the diagonal to ensure that each diagram has the same cardinality; denote this cardinality as $m$. To compute the weighted Fréchet mean, we need to find $\mathbb{M}_k = \{y^{(i)}\}_{i=1}^m$ that minimises the Fréchet function in (2). Implicit to the Wasserstein distance is a bijection $\gamma_j : y^{(i)} \mapsto x_j^{(i)}$ for each $j$. Supposing we know these bijections, we can rearrange the Fréchet function into the form $F(\mathbb{M}_k) = \sum_{j=1}^{n} r_{jk}^2 W_2(\mathbb{M}_k, \mathbb{D}_j)^2 = \sum_{i=1}^{m} \sum_{j=1}^{n} r_{jk}^2 ||y^{(i)} - x_j^{(i)}||^2$. In this form, the summand is minimised for $y^{(i)}$ by the weighted Euclidean centroid of the points $\{x_j^{(i)}\}_{j=1}^n$. Therefore to compute the weighted Fréchet mean, we need to find the correct bijections. We start by using the Hungarian algorithm to find an optimal matching between $\mathbb{M}_k$ and each $\mathbb{D}_j$. Given a $\mathbb{D}_j$, for each point $y^{(i)} \in \mathbb{M}_k$, the Hungarian algorithm will assign an optimally matched point $x_j^{(i)} \in \mathbb{D}_j$. Specifically, we find matched points

$$\left[x_j^{(i)}\right]_{i=1}^m \longleftarrow \text{Hungarian}(\mathbb{M}_k, \mathbb{D}_j), \text{ for each } j = 1, \ldots, n. \tag{3}$$

Now, for each $y^{(i)} \in \mathbb{M}_k$ we need to find the weighted average of the matched points $\left[x_j^{(i)}\right]_{j=1}^n$. However, some of these points could be copies of the diagonal, so we need to consider three distinct cases: that each matched point is off-diagonal, that each one is a copy of the diagonal, or that the points are a mixture of both. We start by partitioning $1, \ldots, n$ into the indices of the off-diagonal points $\mathscr{J}_{\text{OD}}^{(i)} = \left\{j : x_j^{(i)} \neq \Delta\right\}$ and the indices of the diagonal points $\mathscr{J}_{\text{D}}^{(i)} = \left\{j : x_j^{(i)} = \Delta\right\}$ for each $i = 1, \ldots, m$. Now, if $\mathscr{J}_{\text{OD}} = \emptyset$ then $y^{(i)}$ is a copy of the diagonal. If not, let $w = \left(\sum_{j \in \mathscr{J}_{\text{OD}}^{(i)}} r_{jk}^2\right)^{-1} \sum_{j \in \mathscr{J}_{\text{OD}}^{(i)}} r_{jk}^2 x_j^{(i)}$ be the weighted mean of the off-diagonal points. If $\mathscr{J}_{\text{D}}^{(i)} = \emptyset$, then $y^{(i)} = w$. Otherwise, let $w_\Delta$ be the point on the diagonal closest to $w$. Then our update is

$$y^{(i)} \longleftarrow \frac{\sum_{j \in \mathscr{J}_{\text{OD}}^{(i)}} r_{jk}^2 x_j^{(i)} + \sum_{j \in \mathscr{J}_{\text{D}}^{(i)}} r_{jk}^2 w_\Delta}{\sum_{j=1}^{n} r_{jk}^2} \tag{4}$$

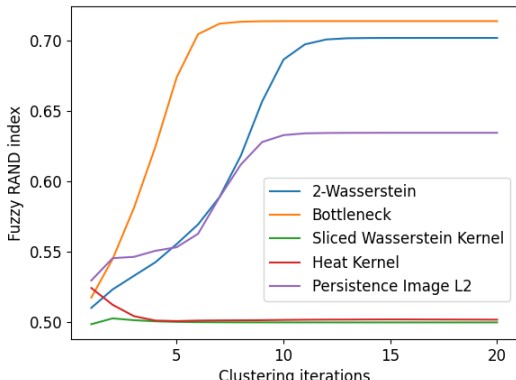

Figure 3: For computational speedups practitioners may wish to use different distances in the clustering algorithm. We use the fuzzy RAND index (Campello, 2007) to evaluate cluster quality when using some common distances. The more expensive optimal matching-based distances perform best, whereas approximations and embedding-based distances are faster but score lower.

for $i = 1, \ldots, m$. We alternate between (3) and (4) until the matching remains the same. Theorem 2, proving that this algorithm converges to a local minimum of the Fréchet function, is proven in Appendix B.

THEOREM 2:
Given diagrams $\mathbb{D}_j$, membership values $r_{jk}$, and the Fréchet function $F(\hat{\mathbb{D}}) = \sum_{j=1}^n r_{jk}^2 W_2(\hat{\mathbb{D}}, \mathbb{D}_j)^2$, then $\mathbb{M}_k = \{y^{(i)}\}_{i=1}^m$ is a local minimum of $F$ if and only if there is a unique optimal pairing from $\mathbb{M}_k$ to each of the $\mathbb{D}_j$ and each $y^{(i)}$ is updated via (4).

## 4. Experiments

### 4.1. Synthetic data

**Example clustering.** We start by demonstrating our algorithm on a simple synthetic dataset designed to highlight its ability to cluster based on topology. We produce three datasets of noise, three datasets of a ring, and three datasets of figure-of-eights, all shown on the left of Figure 1. In the middle of Figure 1 we show the corresponding 1-PH persistence diagrams. Note that the persistence diagrams have either zero, one, or two significant off-diagonal points, corresponding to zero, one, or two holes in the datasets. We then use our algorithm to cluster the nine persistence diagrams into three clusters. Having only been given the list of diagrams, the number of clusters, and the maximum number of iterations, our algorithm successfully clusters the diagrams based on their topology. The right of Figure 1 shows that the cluster centres have zero, one, or two off-diagonal points: our algorithm has found cluster centres that reflect the topological features of the datasets.

    **Empirical behaviour.** Figure 2 shows the results of experiments run to determine the empirical performance of our algorithm. We give theoretical guarantees that every conver-

gent subsequence will tend to a local minimum, but in practice it remains important that our algorithm will converge within a reasonable timeframe. To this end we ran experiments on a total of 825 randomly generated persistence diagrams, recording the number of iterations and cost functions for both the FCM clustering and the weighted Fréchet mean (WFM) computation. We considered the FCM to have converged when the cost function remained within $\pm 0.5\%$. As explained in Section 3.2, the WFM converges when the matching stays the same. Our experiments showed that the FCM clustering consistently converges within 5 iterations, regardless of the number of diagrams and points per diagram (note that the time per iteration increases as the number of points/diagrams increases, even if the number of iterations remains stable). We had no experiments in which the algorithm did not converge. The WFM computation requires more iterations as both number of diagrams and number of points per diagram increases, but we once again experienced no failures to converge.

The use of the Wasserstein distance in the clustering still means that some large-scale problems are computationally intractable. To explore solutions to this, we investigated the use of different distances in Equation (1). Specifically, we evaluated the quality of learnt clusters using the fuzzy RAND index (Campello, 2007) when clustering with the 2-Wasserstein distance, bottleneck distance, sliced Wasserstein kernel (Carrière et al., 2017), heat kernel, and L2 distance between persistence images (Chepushtanova et al., 2015). We find that the more expensive optimal matching-based distances perform best, whereas approximations and embedding-based distances are faster but score lower. These results are shown in Figure 3.

## 4.2. Deep Learning Model Selection

Previous initial work has suggested that deep learning models will perform better on unseen datasets which have a similar topological complexity to the model's decision boundary (Ramamurthy et al., 2019). In fact, there is an increasing amount of work studying the link between topology and neural network performance (Rieck et al., 2018; Guss and Salakhutdinov, 2018). To this end we utilise our algorithm to cluster the topology of the decision boundaries of pre-trained models and tasks (labelled datasets). Given a task we find the nearest cluster centre, then select the models nearest to that centre. Even though the only information utilised for the model selection is the topology of the decision boundaries, we find that it consistently selects the top performing model as the first choice, and additional choices perform above average, despite not being trained on the task. This further confirms that the topology of the decision boundary is indicative of generalisation ability to unseen tasks. Furthermore, our algorithm is able to exploit this information to learn cluster centres that consistently select the best performing models on tasks.

Specifically, given a dataset with $n$ classes, we fix one class to define $n - 1$ *tasks*: binary classification of the fixed class vs each of the remaining classes. On each of these tasks, we train a *model*. We compute the decision boundary of the model $f$, defined as $(x_1, \ldots, x_m, f(x))$ where $f(x)$ is the model's prediction for $x = (x_i)_i$, and the decision boundary of the tasks, defined via the labelled dataset as $(x_1, \ldots, x_m, y)$ where $y$ is the true label. We compute the 1-persistence diagrams of the tasks' and models' decision boundaries and cluster them to obtain membership values and cluster centres. To view task and model proximity through our clustering, we find the cluster centre with the highest membership

value for each task, and consider the models closest to that cluster centre. Note that model selection is naturally a fuzzy task: one model can (and does) perform well on multiple tasks. Therefore this is a task best suited to fuzzy clustering. We further discuss why hard clustering does not work here in Appendix C.3.

To assess the ability of our model/task clustering, we performed the above experiment on three different datasets: MNIST (LeCun et al., 2010), FashionMNIST (Xiao et al., 2017), and Kuzushiji-MNIST (Clanuwat et al., 2018). We repeat each experiment three times using sequential seeds, resulting in a total of 81 trained models. Our goal is to evaluate whether or not the clustering is capturing information about model performance on tasks, so as a baseline we use the average performance of all models on a fixed task, averaged over all tasks. We start by verifying what happens if we use the model closest to the cluster centre associated with the task (i.e., top-1). We see a significant increase in performance, indicating that the topological fuzzy clustering has selected the model trained on the task, despite only having information about the topology of the decision boundary. We also compute the top-3 and top-2 performance change over average. We still see a statistically significant increase in performance over average performance, indicating that the fuzzy clusters are capturing information about model generalisation to unseen tasks. These results are shown in Figure 4.

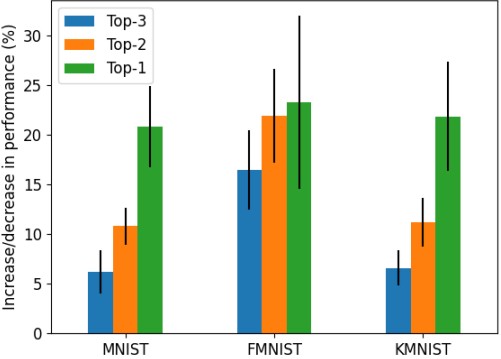

Figure 4: Performance increase/decrease over average task performance when using the fuzzy clustered persistence diagrams of decision boundaries for model selection. Given a task, we find its nearest cluster centre, and use fuzzy membership values to select the nearest models. The increase in performance demonstrates that our fuzzy clustering automatically clusters models near tasks they perform well on, using just the topology of their decision boundaries.

## 5. Conclusion

We have extended Fuzzy c-Means clustering to the space of persistence diagrams, adding an important class of unsupervised learning algorithm to Topological Data Analysis' toolkit. We give theoretical and empirical convergence results, and study applications to model se-

lection in deep learning. We find the results on decision boundaries particularly interesting: successfully matching pre-trained models to new tasks by fuzzy clustering a topological representation of their decision boundaries could be a useful tool in a model marketplace scenario. Future work should further investigate the ability of the topology of the decision boundaries to quantify generalisation to unseen tasks in deep learning models.

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

## Appendix A. Convergence of the FCM clustering algorithm

We first need to consider the update steps (1) and (2) as a single update procedure. Let $F : \mathbb{M} \mapsto R$ be defined by (1) and $G : R \mapsto \mathbb{M}$ be defined by (2), and for $R = \{r_{jk}\}$ and $\mathbb{M} = \{\mathbb{M}_k\}$ consider the sequence

$$\left\{ T^{(l)}(R, \mathbb{M}) : l = 0, 1, \dots \right\}$$

where $T(R, M) = (F \circ G(R), G(R))$. We wish to show convergence of the iterates of $T$ to a local minimum or saddle point of the cost function

$$J(R, \mathbb{M}) = \sum_{j=1}^{n} \sum_{k=1}^{c} r_{jk}^2 W_2(\mathbb{M}_k, \mathbb{D}_j)^2.$$

The two stage update process of $T$ is too complicated to use standard fixed point theorems, so following Bezdek (1980) we shall use the following result, which is proven by Zangwill (1969).

THEOREM 3 (ZANGWILL'S CONVERGENCE THEOREM):
Let $A : X \rightarrow 2^X$ be a point-to-set algorithm acting on $X$. Given $x_0 \in X$, generate a sequence $\{x_k\}_{k=1}^{\infty}$ such that $x_{k+1} \in A(x_k)$ for every $k$. Let $\Gamma \subset X$ be a solution set, and suppose that the following hold.

(i) The sequence $\{x_k\} \subset S \subset X$ for a compact set $S$.

(ii) There exists a continuous function $Z$ on $X$ such that if $x \notin \Gamma$ then $Z(y) < Z(x)$ for all $y \in A(x)$, and if $x \in \Gamma$ then $Z(y) \leq Z(x)$ for all $y \in A(x)$. The function $Z$ is called a descent function.

(iii) The algorithm $A$ is closed on $X \setminus \Gamma$.

Then every convergent subsequence of $\{x_k\}$ tends to a point in the solution set $\Gamma$.

Our algorithm is the update function $T$. We define our solution set as

$$\Gamma = \Big\{ (R, \mathbb{M}) : J(R, \mathbb{M}) < J(\hat{R}, \hat{\mathbb{M}})$$
$$\forall \, (\hat{R}, \hat{\mathbb{M}}) \in B((R, \mathbb{M}), r) \Big\}$$

for some $r > 0$, where the ball surrounding $R$ is the Euclidean ball in $\mathbb{R}^{nc}$ and the ball surrounding $\mathbb{M}$ is $\cup_{k=1}^{c} B_{W_2}(\mathbb{M}_k, r)$. This set contains the local minima and saddle points of the cost function (Bezdek et al., 1987). We wish to show that our cost function $J(R, \mathbb{M})$ is the descent function $Z$. We proceed by verifying each of the requirements for Zangwill's Convergence Theorem.

**Lemma 1** *Every iterate $T^{(l)}(R, \mathbb{M}) \in [0, 1]^{nc} \times \mathrm{conv}(\mathbb{D})^c$, where*

$$\mathrm{conv}(\mathbb{D}) = \bigcup_{k=1}^{c} \bigcup_{\gamma_j} \bigcup_{i=1}^{m} \mathrm{conv}\{\gamma_j(y^{(i)}) : j = 1, \ldots, n\},$$

*with $\gamma_j$ a bijection $\mathbb{M}_k \to \mathbb{D}_j$ and $\mathrm{conv}\{\gamma_j(y^{(i)}) : j = 1, \ldots, n\}$ the ordinary convex hull in the plane. Furthermore, $[0, 1]^{nc} \times \mathrm{conv}(\mathbb{D})^c$ is compact.*

**Proof** By construction, every $r_{jk} \in [0, 1]$. Since $j = 1, \ldots, n$ and $k = 1, \ldots, c$, we can view $R$ as a point in $[0, 1]^{nc}$, and so every iterate of $R$ is in $[0, 1]^{nc}$. We shall show that for a fixed $k$ and a fixed bijection $\gamma_j : \mathbb{M}_k \to \mathbb{D}_j$, each updated $y^{(i)}$ is contained in a convex combination of $\{\gamma_j(y^{(i)}) : j = 1, \ldots, n\}$. Where $\gamma_j(y^{(i)}) = \Delta$, let $\gamma_j(y^{(i)}) = w_\Delta$ as defined in (4), as this is the update point we use for the diagonal. Since there are a finite number of off-diagonal points, each updated $\mathbb{M}_k$ is therefore contained in the union over all bijections and all points $y^{(i)}$ of the convex combination of $\{\gamma_j(y^{(i)}) : j = 1, \ldots, n\}$. By also taking the union over each $k$, we show that every iterate of $\mathbb{M}$ must be contained in the finite triple-union of the convex combination of each possible bijection. To show that each updated $y^{(i)}$ is contained in a convex combination of $\{\gamma_j(y^{(i)}) : j = 1, \ldots, n\}$, recall that $y^{(i)} = \left( \sum_{j=1}^{n} r_{jk}^2 \right)^{-1} \sum_{j=1}^{n} r_{jk}^2 \gamma_j(y^{(i)})$. Letting $t_j^{(i)} = r_{jk}^2 \left( \sum_{j=1}^{n} r_{jk}^2 \right)^{-1}$, clearly each $t_j^{(i)} > 0$ and $\sum_{j=1}^{n} t_j^{(i)} = 1$. Since $y^{(i)} = \sum_{j=1}^{n} t_j^{(i)} \gamma_j(y^{(i)})$, each $y^{(i)}$ is contained in the convex combination. Therefore $T^{(l)}(R, \mathbb{M}) \in [0, 1]^{nc} \times \mathrm{conv}(\mathbb{D})^c$ for each $l = 0, 1, \ldots$.

Now, $[0, 1]$ is closed and bounded, so is compact. The convex hull of points in the plane is closed and bounded, so $\mathrm{conv}\{\gamma_j(y^{(i)}) : j = 1, \ldots, n\}$ is compact. Since finite unions and finite direct products of compact sets are compact, $[0, 1]^{nc} \times \mathrm{conv}(\mathbb{D})^c$ is also compact. ∎

**Lemma 2** *The cost function $J(R, \mathbb{M})$ is a descent function, as defined in Theorem 3(ii).*

**Proof** The cost function $J$ is continuous, as it's a sum, product and composition of continuous functions. Furthermore, we have that for any $(R, \mathbb{M}) \notin \Gamma$,

$$
\begin{aligned}
J(T(R, \mathbb{M})) &= J(F \circ G(R), G(R)) \\
&< J(R, G(R)) \\
&< J(R, M),
\end{aligned}
$$

where the first inequality is due to Proposition 1 in Bezdek (1980), and the second inequality comes from the definition of the Fréchet mean. If $(R, \mathbb{M}) \in \Gamma$ then the strict inequalities include equality throughout. ∎

THEOREM 4:
For any $(R, \mathbb{M})$, every convergent subsequence of $\{T^{(l)}(R, \mathbb{M}) : l = 0, 1, \dots\}$ tends to a local minimum or saddle point of the cost function $J$.

**Proof** We proceed with Zangwill's Convergence Theorem. Our algorithm is the update function $T$, our solution set is $\Gamma$, and our descent function is the cost function $J(R, \mathbb{M})$. By Lemma 4, every iterate is contained within a compact set. By Lemma 5, $J$ is a descent function. Finally, since our function $T$ only maps points in the plane to points in the plane, it is a closed map. The theorem follows by applying Theorem 3. ∎

## Appendix B. Convergence of the Fréchet mean algorithm

Recall that the Fréchet mean is computed by finding the $\arg\min$ of

$$
F(\hat{\mathbb{D}}) = \sum_{j=1}^{n} r_{jk}^2 F_j(\hat{\mathbb{D}}), \text{ with } F_j(\hat{\mathbb{D}}) = W_2(\hat{\mathbb{D}}, \mathbb{D}_j)^2, \tag{5}
$$

for fixed $k$. We start by recounting work by Turner et al. (2012), which this section adapts for the weighted Fréchet mean.[2] The proofs we're adapting use a gradient descent technique to prove local convergence. In order to use their techniques, we need to define a differential structure on the space of persistence diagrams.

By Theorem 2.5 from Turner et al. (2012), the space of persistence diagrams $\mathscr{D}_{L^2} = \{\mathbb{D} : W_2(\mathbb{D}, \Delta) < \infty\}$ is a non-negatively curved Alexandrov space. An optimal bijection $\gamma : \mathbb{D}_1 \to \mathbb{D}_2$ induces a unit-speed geodesic $\phi(t) = \{(1 - t)x + t\gamma(x) : x \in \mathbb{D}_1, 0 \le t \le 1\}$. For a point $\mathbb{D} \in \mathscr{D}_{L^2}$ we define the tangent cone $T_{\mathbb{D}}$. Define $\hat{\Sigma}_{\mathbb{D}}$ as the set of all non-trivial unit-speed geodesics emanating from $\mathbb{D}$. Let $\phi, \eta \in \hat{\Sigma}_{\mathbb{D}}$ and define the angle between them as

$$
\angle_{\mathbb{D}}(\phi, \eta) = \arccos\left(\lim_{s,t\downarrow 0} \frac{s^2 + t^2 - W_2(\phi(s), \eta(t))^2}{2st}\right)
$$

---

2. In Turner et al. (2012), the Fréchet mean is defined as the $\arg\min$ of the Fréchet function $F(\hat{\mathbb{D}}) = \int W_2(\hat{\mathbb{D}}, \mathbb{D}_j)^2 d\rho(\hat{\mathbb{D}})$ with the empirical measure $\rho = n^{-1} \sum_{j=1}^{n} \delta_{\mathbb{D}_j}$. We are using the empirical measure $\rho = \left(\sum_{j=1}^{n} r_{jk}^2\right)^{-1} \sum_{j=1}^{n} r_{jk}^2 \delta_{\mathbb{D}_j}$, but for ease we drop the scalar $\left(\sum_{j=1}^{n} r_{jk}^2\right)^{-1}$ as it is positive, so it does not affect the minimum of the function.

in $[0, \pi]$ when the limit exists. Then the space of directions $(\Sigma_{\mathbb{D}}, \angle_{\mathbb{D}})$ is the completion of $\hat{\Sigma}_{\mathbb{D}}/\sim$ with respect to $\angle_{\mathbb{D}}$, with $\phi \sim \eta \iff \angle_{\mathbb{D}}(\phi, \eta) = 0$. We now define the tangent cone as

$$T_{\mathbb{D}} = (\Sigma_{\mathbb{D}} \times [0, \infty))/(\Sigma_{\mathbb{D}} \times \{0\}).$$

Given $u = (\phi, s), v = (\eta, t)$, we define an inner product on the tangent cone by

$$\langle u, v \rangle = st \cos \angle_{\mathbb{D}}(\phi, \eta).$$

Now, for $\alpha > 0$ denote the space $(\mathscr{D}_{\mathrm{L}^2}, \alpha W_2)$ as $\alpha \mathscr{D}_{\mathrm{L}^2}$ and define the map $i_\alpha : \alpha \mathscr{D}_{\mathrm{L}^2} \to \mathscr{D}_{\mathrm{L}^2}$. For an open set $\Omega \subset \mathscr{D}_{\mathrm{L}^2}$ and a function $f : \Omega \to \mathbb{R}$, the differential of $f$ at $\mathbb{D} \in \Omega$ is defined by $d_{\mathbb{D}} f = \lim_{\alpha \to \infty} \alpha(f \circ i_{\mathbb{D}} - f(\mathbb{D}))$. Finally, we say that $s \in T_{\mathbb{D}}$ is a supporting vector of $f$ at $\mathbb{D}$ if $d_{\mathbb{D}} f(x) \leq -\langle s, x \rangle$ for all $x \in T_{\mathbb{D}}$.

**Lemma 3** *The following two results are proven in* Turner et al. (2012).

(i) *Let* $\mathbb{D} \in \mathscr{D}_{\mathrm{L}^2}$. *Let* $F_j(\hat{\mathbb{D}}) = W_2(\hat{\mathbb{D}}, \mathbb{D}_j)^2$. *Then if* $\phi$ *is a distance-achieving geodesic from* $\mathbb{D}$ *to* $\hat{D}$, *then the tangent vector to* $\phi$ *at* $\mathbb{D}$ *of length* $2W_2(\hat{\mathbb{D}}, \mathbb{D})$ *is a supporting vector at* $\mathbb{D}$ *of* $f$.

(ii) *If* $\mathbb{D}$ *is a local minimum of* $f$ *and* $s$ *is a supporting vector of* $f$ *at* $\mathbb{D}$, *then* $s = 0$.

If there is a unique optimal matching $\gamma_{\mathbb{D}_1}^{\mathbb{D}_3} : \mathbb{D}_1 \to \mathbb{D}_3$, we say that it is induced by an optimal matching $\gamma_{\mathbb{D}_1}^{\mathbb{D}_2} : \mathbb{D}_1 \to \mathbb{D}_2$ if there exists a unique optimal matching $\gamma_{\mathbb{D}_2}^{\mathbb{D}_3} : \mathbb{D}_2 \to \mathbb{D}_3$ such that $\gamma_{\mathbb{D}_1}^{\mathbb{D}_3} = \gamma_{\mathbb{D}_2}^{\mathbb{D}_3} \circ \gamma_{\mathbb{D}_1}^{\mathbb{D}_2}$. Proposition 3.2 from Turner et al. (2012) states that an optimal matching at a point is also locally optimal. In particular, it states the following.

**Lemma 4** *Let* $\mathbb{D}_1, \mathbb{D}_2 \in \mathscr{D}_{\mathrm{L}^2}$ *such that there is a unique optimal matching from* $\mathbb{D}_1$ *to* $\mathbb{D}_2$. *Then there exists an* $r > 0$ *such that for every* $\mathbb{D}_3 \in B_{W_2}(\mathbb{D}_2, r)$, *there is a unique optimal pairing from* $\mathbb{D}_2$ *to* $\mathbb{D}_3$ *that is induced by the matching from* $\mathbb{D}_1$ *to* $\mathbb{D}_2$.

The following theorem proves that our algorithm converges to a local minimum of the Fréchet function.

THEOREM 5:
Given diagrams $\mathbb{D}_j$, membership values $r_{jk}$, and the Fréchet function $F$ defined in (5), then $\mathbb{M}_k = \{y^{(i)}\}_{i=1}^m$ is a local minimum of $F$ if and only if there is a unique optimal pairing from $\mathbb{M}_k$ to each of the $\mathbb{D}_j$, denoted $\gamma_j$, and each $y^{(i)}$ is updated via (4).

**Proof** First assume that $\gamma_j$ are optimal pairings from $\mathbb{M}_k$ to each $\mathbb{D}_j$, and let $s_j$ be the vectors in $T_{\mathbb{M}_k}$ that are tangent to the geodesics induced by $\gamma_j$ and are distance-achieving. Then by Lemma 7(i), each $2s_j$ is a supporting vector for the function $F_j$. Furthermore, $2\sum_{j=1}^n r_{jk}^2 s_j$ is a supporting vector for $F$, as for any $\hat{\mathbb{D}}$,

$$d_{\mathbb{M}_k} F(\hat{\mathbb{D}}) = d_{\mathbb{M}_k} \left( \sum_{j=1}^n r_{jk}^2 F_j(\hat{\mathbb{D}}) \right) = \sum_{j=1}^n r_{jk}^2 d_{\mathbb{M}_k} F_j(\hat{\mathbb{D}})$$

$$\leq \sum_{j=1}^n -r_{jk}^2 \langle 2s_j, \hat{\mathbb{D}} \rangle = -\left\langle 2 \sum_{j=1}^n r_{jk}^2 s_j, \hat{\mathbb{D}} \right\rangle.$$

By Lemma 7(ii), $2 \sum_{j=1}^{n} r_{jk}^2 s_j = 0$. Putting $s_j = \gamma_j(y^{(i)}) - y^{(i)}$ and rearranging gives that $y^{(i)}$ updates via (4), as required. Note that when $\gamma_j(y^{(i)}) = \Delta$, we let $\gamma_j(y^{(i)}) = w_\Delta$ as defined in (4), because this minimises the transportation cost to the diagonal. Now suppose that $\gamma_j$ and $\tilde{\gamma}_j$ are both optimal pairings. Then by the above argument $\sum_{j=1}^{n} r_{jk}^2 s_j = \sum_{j=1}^{n} r_{jk}^2 \tilde{s}_j = 0$, implying that $s_j = \tilde{s}_j$ and so $\gamma_j = \tilde{\gamma}_j$. Therefore the optimal pairing is unique.

To prove the opposite direction, assume that $\mathbb{M}_k = \{y^{(i)}\}$ locally minimises the Fréchet function $F$. Observe that for a fixed bijection $\gamma_j$, we have that

$$F(\mathbb{M}_k) = \sum_{j=1}^{n} r_{jk}^2 W_2(\mathbb{M}_k, \mathbb{D}_j)^2$$

$$= \sum_{j=1}^{n} r_{jk}^2 \left( \inf_{\gamma_j : \hat{M} \to \mathbb{D}_j} \sum_{y \in \mathbb{M}_k} ||y - \gamma_j(y)||^2 \right)$$

$$= \sum_{j=1}^{n} r_{jk}^2 \sum_{i=1}^{m} ||y^{(i)} - x_j^{(i)}||^2$$

$$= \sum_{i=1}^{m} \left( \sum_{j=1}^{n} r_{jk}^2 ||y^{(i)} - x_j^{(i)}||^2 \right).$$

The final term in brackets is non-negative, and minimised exactly when $y^{(i)}$ is updated via (4). Furthermore, the unique optimal pairing from $\mathbb{M}_k$ to each of the $\mathbb{D}_j$'s is the same for every $\hat{M}$ within the ball $B_{W_2}(\mathbb{M}_k, r)$ for some $r > 0$, by Lemma 8. Therefore, if $\mathbb{M}_k$ is a local minimum of $F$, then the $y^{(i)}$'s are equal to the values found by taking the optimal pairings $\gamma_j$ and calculating the weighted means of $\gamma_j(y^{(i)})$ with the weights $r_{jk}^2$, as required. It will remain a minimum as long as the matching stays the same, which happens in the ball $B_{W_2}(\mathbb{M}_k, r)$, so we are done. ∎

## Appendix C. Experimental details

### C.1. Synthetic data

**Membership values.** The membership values for the synthetic datasets are in Table 1. Datasets 1-3 are the datasets of noise, datasets 4-6 are the datasets with one ring, and datasets 7-9 are the datasets with two rings. We ran our algorithm for 20 iterations. The code to generate the dataset is available in the supplementary materials.

Table 1: Membership values for the synthetic dataset

| Dataset | 1 | 2 | 3 | 4 | 5 | 6 | 7 | 8 | 9 |
|---|---|---|---|---|---|---|---|---|---|
| Cluster 1 | 0.6336 | 0.5730 | 0.5205 | 0.2760 | 0.2503 | 0.1974 | 0.2921 | 0.2128 | 0.2292 |
| Cluster 2 | 0.1768 | 0.2057 | 0.2327 | 0.5361 | 0.5329 | 0.6371 | 0.2452 | 0.2291 | 0.1822 |
| Cluster 3 | 0.1900 | 0.2212 | 0.2468 | 0.1879 | 0.2169 | 0.1655 | 0.4627 | 0.5580 | 0.5885 |

Table 2: Seconds per clustering iteration

| Points | 100 | 200 | 300 | 400 | 500 | 600 | 700 | 800 | 900 | 1000 |
|---|---|---|---|---|---|---|---|---|---|---|
| **FPDCluster** | **0.01552** | **0.1975** | **0.9358** | **2.229** | **5.694** | **12.29** | **19.27** | **34.50** | **53.20** | **77.81** |
| ADMM | 5.622 | 34.86 | 161.3 | 617.6 | - | - | - | - | - | - |
| BADMM | 0.2020 | 2.188 | 26.38 | 112.6 | - | - | - | - | - | - |
| SubGD | 0.4217 | 2.273 | 22.17 | 103.4 | - | - | - | - | - | - |
| IterBP | 0.3825 | 2.226 | 21.57 | 108.9 | - | - | - | - | - | - |
| LP | 0.3922 | 2.031 | 22.32 | 117.3 | - | - | - | - | - | - |

**Timing experiments.** For the timing experiments we divide the total number of points equally between four distributions, two of which are noise and two of which are shaped in a ring. Each clustering algorithm was run for five iterations on one core of a 2018 MacBook Pro with a 1.4GHz Intel Core i5. We included the time taken to compute the persistence diagrams in the running times for our algorithm.

We also use synthetic data to empirically compare the running time of our algorithm to other dataset clustering algorithms available. Computing the Wasserstein distance has super-cubic time complexity Ling and Okada (2007), so is a significant bottleneck both for our algorithm and comparable Wasserstein barycentre clustering algorithms Benamou et al. (2015); Cuturi and Doucet (2014); Li and Wang (2008); Ye and Li (2014); Ye et al. (2017). Persistence diagrams generally reduce both the dimensionality and number of points in a dataset,[3] so we in turn reduce the computational bottleneck. To demonstrate this, we evaluated the average time per iteration of our persistence diagram clustering algorithm, as well as the average iteration time for comparable Wasserstein barycentre clustering algorithms. We included the time taken to compute the persistence diagrams from the datasets when timing our clustering algorithm. We give the results in Table 2, leaving an entry blank where it became unpractical to run a test (e.g. it takes too long to return a solution and the algorithm becomes unresponsive). We show at least an order of magnitude improvement in performance over comparable Wasserstein barycentre clustering algorithms.

**Empirical performance experiments.** We get empirical results on the convergence of a total of 825 randomly generated persistence diagrams. Following Euclidean fuzzy clustering, we denote convergence when the cost function is stable to within $\pm 0.5\%$. The WFM converges when the matching remains stable, which we proves does happen. We use the seeds 0, 1, and 2 respectively for our repeats.

We implement the Fuzzy RAND index (Campello, 2007), available in the supplementary materials. We use the same synthetic dataset as before to evaluate our cluster quality, with the origin of the data (noise, one hole, or two holes) as a reference partition. We used Persim[4] to compute the additional distances.

---

3. Persistence diagrams are always planar, so if the data is in $\mathbb{R}^d, d > 2$, then there is a dimensionality reduction. For $p > 0$, the persistence diagram of $p$-PH always has less points than the dataset when computed with the Vietoris-Rips complex.

4. https://persim.scikit-tda.org/en/latest/

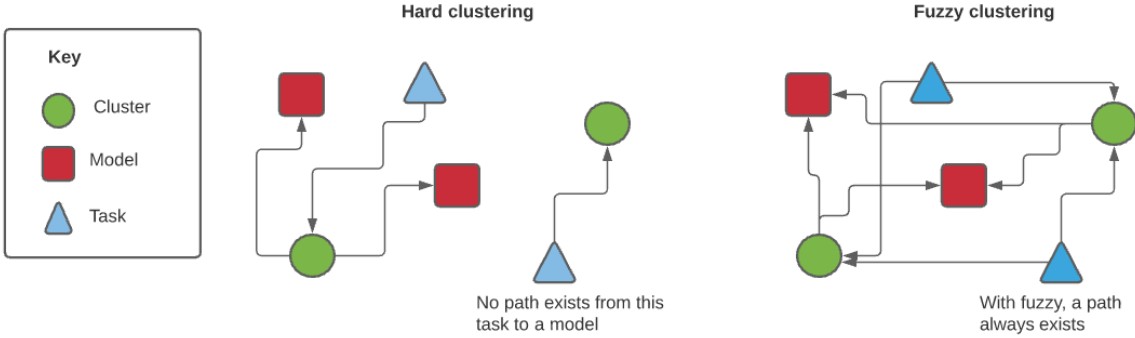

Figure 5: With hard clustering, we cannot always find a path from a task to a model.

## C.2. Decision boundaries

**Why hard clustering does not work.** In order to assign each task to the top-ranked models, we need to have a path from a task to the nearest cluster centre, then from that cluster centre to the $k$-nearest models (note that when we refer to models/tasks, we're implicitly referring to the persistence diagram of their decision boundary). We can always find that route when fuzzy clustering, as the fractional membership values mean that we have information about the proximity of every model/task with every cluster centre. However, with hard clustering we cannot always find that route. Firstly, the hard labelling means that you lose a lot of information about the proximity of models/tasks to cluster centres. Therefore, in order to find a route, we need a every task to be assigned to a cluster centre that also has a model assigned to it. However, there are no guarantees that will happen. We show an example where no path exists in Figure 5.

**Experimental details.** All code used for computation is available in the supplementary materials. For models, we trained the standard Pytorch CNN available at https://github.com/pytorch/examples/blob/master/mnist/main.py. We trained them on MNIST, FashionMNIST, and KMNIST, each obtained using the Torchvision.datasets package. We split the data into 9 binary datasets for classification, class 0 vs each of the remaining classes. We trained three of each model, seeded with 0, 1, and 2 respectively. MNIST and KMNIST were each trained for five epochs, FashionMNIST was trained for 14 epochs. Our train:test split was 6:1, as is standard for MNIST structured datasets. We used Ripser to compute the 1-persistence diagrams using default settings. We limited the number of points in the diagram to the 25 most persistent when clustering. Our percentage improvement values use the membership values after 16 iterations. We compute the standard error bounds when calculating the percentage improvement.

