# OpenReview forum: "Fuzzy c-Means Clustering in Persistence Diagram Space for Deep Learning Model Selection"
_NeurIPS.cc/2022/Workshop/NeurReps — NeurReps 2022 Poster_

### Official Review · Reviewer_RMT1 · 2022-10-13

**Confidence:** 2
**Soundness:** 4
**Presentation:** 3
**Contribution:** 3
**Overall Rating:** 7

**Summary:**

The paper extends the classic Fuzzy c-means clustering algorithm to the space of persistence diagrams known from topological data analysis. This is realized by placing a Wasserstein metric over the space of persistence diagrams as, at least for small problems, it is feasible to compute Frechet averages under this metric. Convergence guarantees are provided that match the classic Euclidean results. Empirical results on small tasks demonstrate that the method performs as expected.

**Questions:**

See above.

**Limitations:**

I think the paper could have been more clear on the limitations of the work.

**Recommended Decision:**

3: Accept

**Relevance:**

4: Highly relevant

**Strengths And Weaknesses:**

Strengths:
* The paper is very clear to read.
* The proposed generalization of FCM is, to the best of my knowledge, novel.
* From a theoretical perspective, the work is fairly complete as the generalization seems to have similar properties to that of the original Euclidean formulation. This is as much as we can realistically ask for.

Weaknesses:
* I do not quite understand why it is interesting to cluster according to topology. I can conceptually see that it can be relevant to group datasets according to the topology of the underlying data manifold, but is a single persistence diagram enough to reach that goal? It would be helpful with a motivation that speaks to the practitioner.
* I didn't quite understand the model selection problem being tackled in Sec. 4.2.

Minor:
* On page 2, the citation for the Sinkhorn algorithm should use \citep. Also, is that the appropriate reference?
* In Fig 1, it would be nice with axis labels on the diagrams. Also, I never managed to figure out how to interpret the right panel. A bit of hand-holding would be valuable here.
* After Eq. 4 there is an unwanted indentation caused by a linebreak too many.

**Submission Track:**

Proceedings Paper (9 Page)

---

### Official Review · Reviewer_wLRC · 2022-10-14
**Relevant application but most of the contributions were the subject of a previous publication**

**Confidence:** 4
**Soundness:** 3
**Presentation:** 3
**Contribution:** 2
**Overall Rating:** 5

**Summary:**

This paper proposes to extend the Fuzzy $c$-Means algorithm to the space of persistence diagrams and illustrates the method on the problem of pre-trained model selection in deep learning. The general idea is to rewrite the Fuzzing $c$-Means algorithm using the Wasserstein distance on the space of persistence diagrams instead of the Euclidean distance. This new method allows to perform topology-aware fuzzy clustering. Typically, an experience run by the authors shows that it is able to cluster some points cloud based on the genus of the surface they are sampled from. The authors also show that the convergence theoretical guarantees of the traditional Fuzzy $c$-Means algorithm still hold in their case. Then they derive a method for selecting the right model to perform a classification task. The problem of model selection for given tasks in deep learning is by nature a fuzzy problem as a model can perform well on multiple tasks. As Ramamurthy et al. showed that deep learning models perform better on tasks if they have topologically similar decision boundary, the authors propose to perform fuzzy clustering on pre-trained models and tasks as represented by the persistence diagrams of their decision boundary. The numerical results show that not only does the clustering successfully retrieve the corresponding pre-trained model for each task (i.e. it clusters this model as the closest to the task) but also provide other good candidates to perform well on the task even when they have not been trained (i.e. using one of the closest models to a given task significantly increase the performance compared to average).

**Questions:**

One question about the algorithmic design. Using the notations of Bezdek, 1984, what is the value of $m$ you chose in your extension of the method ? According to formula (2), I infer than $m=2$, but this is not coherent with formula (1) with suggests that $m=3$ in order to have $2/(m-1)=1$. I imagine that the results and the proof only hold for a common value of $m$.

Suggestion. I believe that the paper would significantly gain in originality from focusing on the problem of model selection as an application of the method. To be fully considered as an original contribution, the paper needs to be restructured with this point of view and enriched with applications and experiments, while referring to the previous paper for all the details on the method and the theoretical results together with proofs . The application on model selection can be studied and further, and it would also be interesting to see experiments on other datasets showing both the relevance of fuzzy clustering and topology (for example, where the use of topological features allows to replace supervised learning by unsupervised learning).

**Limitations:**

The average computational cost of the method has been the studied, and the authors also compare quality of clustering using different methods which arise in topological data analysis, highlighting some compromise between accuracy and speed of convergence. It could also be interesting to compare different computations of the Fréchet mean for the Wasserstein distance.

**Recommended Decision:**

1: Reject

**Relevance:**

4: Highly relevant

**Strengths And Weaknesses:**

The method is clearly presented and well motivated by the significant illustration. Indeed the application tackles one of the challenging questions encountered in classification, i.e. the choice of a model for a given classification task, and the experiment shows that the method performs quite well on this problem . The paper also provide with theoretical results on the convergence of the algorithm, derived from the ones of the original Fuzzy $c$-Means paper.

However, while the application is as far as I know innovative and particularly relevant for the method, the method itself and its guarantees were already introduced in a previous version of the paper in 2020, which has been presented at the Neurreps workshop on Topological Data Analysis, and implemented in the context of the ICLR 2021 computational geometry and topology challenge. Therefore, this part of the contribution can not be completely considered as a novel contribution.

**Submission Track:**

Proceedings Paper (9 Page)

---

### Official Review · Reviewer_4nLt · 2022-10-17
**Interesting method with limited empirical results**

**Confidence:** 4
**Soundness:** 3
**Presentation:** 3
**Contribution:** 2
**Overall Rating:** 5

**Summary:**

This paper extends the ubiquitous Fuzzy c-Means (FCM) clustering algorithm to the space of persistence diagrams, enabling unsupervised learning in a topological setting. The paper gives theoretical convergence guarantees that correspond to the Euclidean case and empirically demonstrate the capability of the clustering to capture topological information via the fuzzy RAND index. This paper also presents an application of the algorithm to a scenario that utilises both the topological and fuzzy nature of the proposed algorithm: pre-trained model selection in deep learning.

**Questions:**

None

**Recommended Decision:**

2: Borderline

**Relevance:**

3: Solid fit

**Strengths And Weaknesses:**

**Strengths**:

1. The paper is well-written.
2. The paper provides proof of the convergence of the algorithm.

**Weaknesses**:

The experiments are relative limited. Only three small datasets are tested in this work.

**Submission Track:**

Proceedings Paper (9 Page)

---

### Decision · Program_Chairs · 2022-10-21

Accept (Poster)